# How Do Farmers Respond to Water Resources Management Policy in the Heihe River Basin of China?

**Guifang Li [1], Dingyang Zhou [2] and Minjun Shi [3],***

[1] School of Economics, Renmin University of China, Beijing 100872, China; liguifang55@163.com

[2] School of Natural Resources, Faculty of Geographical Science, Beijing Normal University, Beijing 100875, China; zhoudy@bnu.edu.cn

[3] School of Public Affairs, Zhejiang University, Hangzhou 310058, China

* Correspondence: mjshi@zju.edu.cn; Tel./Fax: +86-571-5666-2060

**Abstract:** Reducing agricultural water use is an inevitable choice to alleviate water shortage in arid and semi-arid regions, and high-efficiency irrigation technologies provide conditions for water conservation. However, without unified water resources management policy to redistribute the saved agricultural water, farmers' behavior will lead to water rebound and large-scale expansion of cultivated areas, especially on the edge of oasis regions. To solve these issues and promote the sustainable development of water resources, it makes sense to explore the impact of unified water resources management policy from the perspective of farmers' behavior. This study takes the typical irrigation zone in the Heihe River Basin as a case to discuss the response of farmers' economic behavior to transferring irrigation water and restricting land reclamation, i.e., the unified water resources management policy with the technical efficiency of crop irrigation improved based on the bio-economic model. The results show that in the case of loosening land constraints, farmers will reuse all the saved water for agricultural production by reclaiming unused land or increasing the area of water-intensive crops (vegetables). Although the policy of restricting land reclamation can restrict land expansion, it cannot avoid water rebound caused by adjusting the crop-planting structure. Farmers' land-expansion behavior can be largely restricted by transferring the saved irrigation water to non-agricultural sectors in irrigation zones with inadequate water, but to contain land-expansion behavior in irrigation zones with surplus water, the policy of restricting land reclamation must be implemented simultaneously. The study also reveals that farmers will choose to grow more cash crops (seed maize, vegetables, tomato, seed watermelon, potato, and rapeseed) and fewer food crops (wheat, maize) to increase the profit per unit of water in the scenario of loosening land constraints or transferring agricultural water. Furthermore, the study indicates that farmers' economic income can be decreased or at least not increased with the transfer of agricultural water. Both benefit compensation from non-agricultural sectors and increased non-agricultural income can compensate farmers' economic loss. Therefore, it is necessary to improve water rights trading systems and increase employment opportunities for surplus agricultural labor to promote economic development in rural areas.

**Keywords:** farmers' economic behavior; technical efficiency of crop irrigation; water resources management policy; bio-economic model; Heihe River Basin

## 1. Introduction

Agriculture is the largest water user in many parts of the world with low water use efficiency [1]. This is particularly true in arid and semi-arid regions of China, where oasis agriculture is the leading

industry. Due to widespread increases in population, industrialization, and urbanization, the growing conflicts over water reallocation between water users have become increasingly prominent [2]. Water reallocation from upstream to downstream regions has been implemented in some river basins to alleviate the ecological deterioration in the downstream, but the total use of agricultural water is still increasing in the midstream [3]. Studies show that high-efficiency irrigation technologies can improve agricultural water use efficiency and provide the prerequisite for saving agricultural water [4]. However, such technologies do not lead to an automatic decrease in water use. Farmers' behaviors, driven by pursuing the maximum profit, such as expanding cropland area, increasing irrigation water per acre and growing water-intensive crops [5–7], lead to the reuse of the agricultural water saved by high-efficiency irrigation technologies in the large-scale expansion of agricultural production [8]. A concept used in energy studies, i.e., the "Jevens Paradox" [9] or "Khazzoom-Brookes" hypothesis [10,11], can help us more clearly understanding the phenomenon of water rebound effects [12–14]. The rapid expansion of an oasis is always accompanied by changes in the hydrology process, soil, and ecosystem stability [15,16]. Excessive expansion may increase the high risk of environmental degradation in inland river basins [17]. Hence, farmers' economic behavior not only reduces the effectiveness of policies but also destroys the eco-environment.

These issues have attracted the attention of the government and scholars [18]. In recent years, studies have mainly focused on the connotation, rebound degree, driving factors and avoidance measures of water rebound [19]. The ways to avoid water rebound include restricting the size of the irrigated area, reducing farmers' water use rights, re-assigning the water savings [20,21], and improving water resources management systems [22,23]. The core idea of these policies is to transfer the saved agricultural water to non-agricultural sectors and restrict land reclamation through administrative means. However, few studies explore the impact of these policies from the perspective of farmers' behavior. Farmers are both the users of water resources and the executors of water resources management policy. Changes in farmers' economic behavior not only determine the reallocation of water resources among sectors directly but also affect the timeliness of the policy. Hence, it makes sense to explore farmers' economic behavior in response to the transfer of agricultural water and restricted land reclamation.

Moreover, the level of economic development is generally backward in the inland river basins, and farmers who make a living from agriculture account for up to 70% of all farmers. Scarce available water and land resources are key factors for agricultural production. Related studies reveal that the transfer of saved agricultural water is likely to occur only when farmers are unlikely to explore the opportunity cost of using water in the immediate future [24], and tiered prices based on the value of each crop's marginal product are suggested as an efficient pricing method to enhance the feasibility and effectiveness of policies [25]. In addition, high dependence of household income on agriculture makes farmers reluctant to transfer saved agricultural water, and the problem of surplus rural labor may lead to social instability [26]. Therefore, the policies transferring agricultural water and restricting land reclamation policy must account for regional economic development.

The Heihe River Basin (HRB), the second largest inland river basin in the arid region of northwest China, has faced water conflicts between economic development and eco-environmental services. The oasis in the midstream of the HRB has become an important grain and vegetable commodity production base with a long agricultural development history in northwest China, and the proportion of agricultural water consumption has been approximately 90% for a long time. The pilot project "water-saving society" was promulgated by the Ministry of Water Resources in 2002, aimed at improving water efficiency and promoting water rights trading. The main content included establishing an innovative system for the allocation and trading of water resource property rights, adjusting crop-planting structure and strengthening infrastructure construction [27,28]. In recent years, the technical progress of crop irrigation has been greatly improved in the midstream of the HRB. However, the cropland area increased from 183,620 hm$^2$ to 288,967 hm$^2$ during 2000–2017, and the total agricultural water use has continued to grow over the past few decades. Furthermore, water

rights trading has not been widely applied in many regions because the previous water resources management policy has not been strictly implemented. Many studies show that price control and quota control are the two main water demand management strategies [29] and the public/community involvement is crucial for sustainable water resource management [30]. However, farmers are not very responsive to changes in water price because water prices are currently far below the shadow price of water resources [31]. In comparison, quota control is more effective at reducing agricultural water use by the supporting of water-saving irrigation technology [8,32]. These provide important references for this study, but few studies focus on circumventing the negative effects of high-efficiency irrigation technologies from the perspective of farmers' behavior to promote the coordinated development of water resources and regional economy.

The bio-economic model (BEM) is a comprehensive model combining farmers' economic behavior with agricultural systems' biophysical processes [33–35] and can be used to simulate the impact of agricultural technology progress, agricultural policy adjustment, and market changes on farmers' welfare, agricultural production and the rural eco-environment [36]. This study aims to assess the response of farmers' economic behavior to water resources management policy with the improvement of the technical efficiency of crop irrigation (TECI) and designs the simulation scenario, policy scenario, and compensation scenario to reflect the dynamic link between farmers' economic behavior and policy changes. Compared with the econometric models (Probit model [37], Logit model [38] and Tobit model [39]), the BEM has been widely used in many fields because it can simulate the details of agricultural production activities and is highly sensitive to changes in external factors [40–42].

The remainder of this study is organized as follows. Section 2 introduces the study area; Section 3 builds the BEM model and designs the scenarios; and Section 4 describes and analyzes the response of farmers' economic behavior in different scenarios. Sections 5 and 6 provide some discussions and conclusions.

## 2. Study Area

The HRB originates from the Qilian Mountains in Qinghai province, passes through Gansu province, and ends in East Juyanhai Lake in Inner Mongolia. It covers an area of approximately 130,000 km$^2$. The midstream and part of the upstream region of the HRB are in Zhangye City of Gansu Province, and the downstream region is in the Ejin Banner of Inner Mongolia. Zhangye is an irrigation agriculture economic zone that consists of six counties: Ganzhou and Linze in the Plain Irrigation Zone (PIZ), Minle and Shandan in the Mountain Irrigation Zone (MIZ), and Gaotai in the Northern Desert Irrigation Zone (NDIZ) (Figure 1). The PIZ is in the core area of the oasis, with 142,328 hm$^2$ of unused land, while the NDIZ is located on the edge, with 61,293 hm$^2$ unused land in 2013. The proportion of land outflow in the MIZ is as high as 60%. Unused land makes land reclamation possible. The population, per capita net income and proportion of non-agricultural labor are significantly higher in the PIZ than in the other irrigation zones because the PIZ has abundant land and water resources and relatively ideal irrigation conditions. Due to differences in natural conditions, the crop-planting structure is obviously different in different irrigation zones. Seed maize, maize, and vegetables are mainly grown in the PIZ, cotton, maize-wheat inter-crop and seed watermelon in the NDIZ, and wheat, potato, maize and barley in the MIZ [43] (Tables 1 and 2).

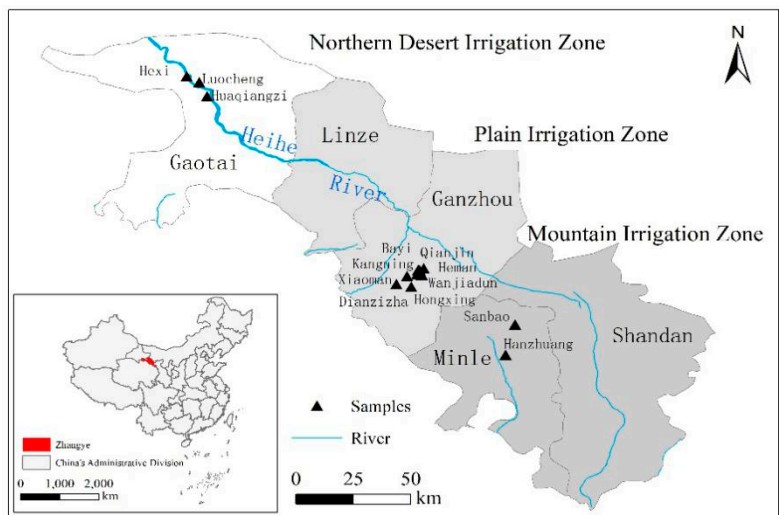

**Figure 1.** Survey site and sample distribution.

**Table 1.** Basic information of typical irrigation zones in Zhangye City in 2013.

| Index | Unit | GDPIZ | GYPIZ | GLNDIZ | MYMIZ |
|---|---|---|---|---|---|
| Population | Person | 104,366 | 91,157 | 12,986 | 72,450 |
| Total labor | Person | 48,070 | 62,217 | 7705 | 41,533 |
| The proportion of non-agricultural labor | % | 15.74 | 16.73 | 11.99 | 12.05 |
| Per capita net income | Yuan/per | 8759 | 9173 | 7805 | 8220 |
| The proportion of agricultural income | % | 53.76 | 59.83 | 79.16 | 34.83 |
| Cultivated area | hm$^2$ | 19,262 | 8322 | 2192 | 17,917 |
| Unused land | hm$^2$ | 142,328 | | 61,293 | 33,333 |
| Surface water | 108 m$^3$ | 1.36 | 1.04 | 0.20 | 0.39 |
| Groundwater | 108 m$^3$ | 0.47 | 0.42 | 0.07 | 0.07 |

**Table 2.** The cropland area and net income of per unit water of main crops in 2013.

| Irrigation Zone | Crop | Cropland Area (hm$^2$) | The Net Income of Per Unit Water (yuan/m$^3$) |
|---|---|---|---|
| PIZ | Seed maize | 12,072 | 1.99 |
| | Maize (mono-crop) | 679 | 1.32 |
| NDIZ | Cotton | 344 | 1.81 |
| | Seed watermelon | 289 | 8.08 |
| | Maize-wheat (inter-crop) | 607 | 2.55 |
| MIZ | Wheat | 3493 | 1.80 |
| | Potato | 2473 | 5.94 |
| | Barley | 573 | 1.43 |
| | Maize (mono-crop) | 106 | 1.59 |

Research shows that there is a certain space to improve the TECI in typical irrigation zones. It is possible to save irrigation water (Table 3) [44]. The process of improving the TECI is dynamic. For example, for seed maize, the irrigation technical efficiency is 0.6553, and the average value of irrigation water is 12,108 (m$^3$/hm$^2$). If the irrigation technical efficiency increased by 20%, the TECI would be 0.7243, the average value of irrigation water would become 11,273 (m$^3$/hm$^2$), and 835 m$^3$ irrigation water could be saved. Farmers' production decisions will change if the saved water is not managed.

Ganzhou, Minle, and Gaotai are the concentration areas of oasis agriculture, with 67.72% of the cultivated area and 75% of the agricultural water. This study takes the Ganzhou Daman irrigation zone

in the PIZ (GDPIZ), Ganzhou Yingke irrigation zone in the PIZ (GYPIZ), Gaotai Luocheng irrigation zone in the NDIZ (GLNDIZ) and Minle Yimin irrigation zone in the MIZ (MYMIZ) as examples to explore farmers' behavior in response to water resources management policy with the improvement of the TECI.

**Table 3.** Changes in the average value of irrigation water with the improvement of the TECI.

| Irrigation Zones | Crop | TECI | Average of Actual Irrigation Water (A) (m³/hm²) | If TECI Increased by 20% (B1) (m³/hm²) | If TECI Increased by 50% (B2) (m³/hm²) | If TECI Increased by 80% (B3) (m³/hm²) | If TECI Increased by 100% (B4) (m³/hm²) |
|---|---|---|---|---|---|---|---|
| PIZ | Seed maize | 0.6553 | 12,108 | 11,273 | 10,021 | 8769 | 7934 |
| | Maize (mono-crop) | 0.6185 | 10,816 | 9991 | 8753 | 7515 | 6690 |
| NDIZ | Cotton | 0.5158 | 6485 | 5857 | 4915 | 3973 | 3345 |
| | Seed watermelon | 0.6518 | 6570 | 6111 | 5423 | 4735 | 4276 |
| | Maize-wheat (inter-crop) | 0.7701 | 6615 | 6310 | 5854 | 5398 | 5094 |
| MIZ | Wheat | 0.8552 | 6471 | 6284 | 6003 | 5722 | 5534 |
| | Potato | 0.6925 | 6261 | 5876 | 5298 | 4720 | 4335 |
| | Barley | 0.7450 | 6255 | 5936 | 5457 | 4979 | 4660 |
| | Maize (mono-crop) | 0.6404 | 7762 | 7204 | 6366 | 5529 | 4971 |

## 3. Methodology and Data

### 3.1. Bio-Economic Model

This study develops an irrigation zone level static 1-year BEM model based on the linear optimization programming structure. This model links household economic activities to crop production activities [31]. The objective function represents the maximization of farmers' net income, and it consists of household production (crop and livestock), household consumption, non-agricultural income, employment cost subsidy, loans, and interest. In the BEM model, production technologies are assumed to be fixed, and the available farming activities are not changed; thus, the Leontief functional form is used for the objective function. The constraint functions include available agriculture acreage, available water resources, labor, nutrition, funds, agricultural production technology, irrigation technology, and so on. The decision variables include the area cultivated for each crop, number of each kind of livestock and quantity of resources assigned to each kind of crop and livestock [4]. The mathematical forms are provided below:

$$
\begin{aligned}
MaxM = &\sum_{c=1}^{C}\left\{P_c\left(\sum_{g=1}^{G}A_{cg}y_{cg}(x)-b_c-s_c\right)-\sum_{g=1}^{G}\sum_{i=1}^{n}A_{cg}e_{icg}x_{icg}\right\}+ \\
&\sum_{v=1}^{V}\left\{P_v(L_vY_v(x)-b_v-s_v)-\sum_{i=1}^{n}e_{iv}x_{iv}\right\}-\sum_{j=1}^{J}P_jf_j+\sum_{o}^{O}w_oz_o-\sum_{k}^{K}w_kh_k
\end{aligned}
\tag{1}
$$

Subject to:

$$
A\geq\sum_{c}^{C}\sum_{g}^{G}A_{cg}+A_r,
\tag{2}
$$

$$
Z_h=z_f+z_o,
\tag{3}
$$

$$
Z_f=z_f+\sum_{k}^{K}h_k,
\tag{4}
$$

$$
\sum_{v=1}^{V}365\alpha_vL_v\leq A_ry_r+T,
\tag{5}
$$

$$365\gamma H \leq \sum_{c=1}^{C} \beta_c b_c + \sum_{j=1}^{J} \beta_j f_j, \tag{6}$$

$$\sum_{c=1}^{C} \sum_{g=1}^{G} \sum_{i=1}^{n} A_{cg} e_{icg} x_{icg} + \sum_{v=1}^{V} \sum_{i=1}^{n} L_v e_{iv} x_{iv} + \sum_{j=1}^{J} P_j f_j + \sum_{k}^{K} w_k h_k \leq M_0 + N + S_s, \tag{7}$$

$$w_{total} \leq (w_s - inf - et) \times cf + w_g, \tag{8}$$

$$\sum_{c=1}^{C} \sum_{g=1}^{G} A_{cg} Q_{cg} + \sum_{v=1}^{V} L_v Q_v \leq w_{total}, \tag{9}$$

The definitions of the variables in the BEM model are given in Table 4. The constraints of resource endowment include land area (Equation (2)), family labor (Equations (3) and (4)), livestock feed requirement (Equation (5)), nutritional requirement (Equation (6)), capital (Equation (7)), water resources (Equations (8) and (9)) and crop rotation.

**Table 4.** Variable definitions.

| Variables | Explanation | Variables | Explanation |
|---|---|---|---|
| $M$ | Net income | $M_0$ | Cash income in the base year |
| $P$ | Price of crop output, livestock output, or purchased food | $c$ | Crop |
| $A$ | Land endowment | $g$ | Land type of cultivated land |
| $y$ | The level of output of crop c or livestock v | $A_{cg}$ | Area of crop c produced on land type g |
| $v$ | Livestock | $x$ | A vector of inputs used in production of crop c or livestock v |
| $s$ | Crop or livestock output y used for self-supply, such as seed, feed, draft animal | $b$ | Crop or livestock output y used for self-consumption |
| $e_i$ | Per unit input cost for input xi | $i$ | Type of vectors of input x |
| $L_v$ | Stock level of livestock v | $j$ | Type of purchased food |
| $w_o$ | Wage for off-farm labor | $f$ | Purchased food |
| $w_k$ | Wage for hired labor | $z_o$ | Family labor used off-farm |
| $A_r$ | Area of range land | $h_k$ | Hired labor used on-farm |
| $z_f$ | Family labor used on-farm | $Z_h$ | Total family labor |
| $\alpha$ | Daily fodder requirement of livestock v | $Z_f$ | Total farm labor input |
| $T$ | Supplementary fodder from crop residue | $y_r$ | Grass yield of range land |
| $H$ | Human population | $\gamma$ | Daily nutrition requirement of human |
| $M_0$ | Cash income in the base year | $\beta$ | Nutrition content of food |
| $S_s$ | Subsidy | $N$ | Amount of available loan |
| $w_{total}$ | Total available water resources | $Q_v$ | Water quota of livestock v |
| $w_g$ | Available groundwater resources | $w_s$ | Allocated surface water based on quota |
| $et$ | Evaporation of runoff | $inf$ | Infiltration of mainstream |
| $Q_{cg}$ | Water quota of crop c on land type g | $cf$ | Canal use efficiency coefficient |

### 3.2. The Farmer Survey

The farmer survey allowed us to gain a better understanding of household economic activities. A quantitative semi-structured survey was undertaken in April and September 2014 in the middle area of the HRB. The survey focused on the family characteristics, costs, and prices of each crop and livestock, land-use types, crop irrigation types, cultivated land use, consumption behavior, etc. The surveyed villages were selected according to the input and output characteristics of the main crops. Under the principle of comprehensive coverage of the sample villages, a total of 1402 interviews were conducted in 13 villages, 4 towns, and 3 counties.

Crop costs mainly included land, labor, capital (seeds, organic materials, chemical fertilizers, pesticides, machinery and fuel costs, irrigation costs), irrigation water, crop yield, and price. We estimated and ranked the average annual cost of each crop. Livestock costs mainly included the quantity and prices of feed and selling price of different types. Feed was the link between crop and livestock. In addition, some qualitative data (such as water supply, consumption preferences and institutional changes) were collected to form a parameter set.

### 3.3. Scenario Design

In this study, three scenarios are designed in Table 5. The first is the simulation scenario, which involves loose land constraints. The second is the policy scenario, which includes three situations: only strengthening land resources management, only strengthening water resources management, and strengthening land and water resources management simultaneously. The final scenario is the compensation scenario, which explores ways to reduce the adverse impact on farmers' production decisions. There are two main situations: in the first, non-agricultural sectors pay for saved agricultural water, and in the other, non-agricultural income is increased [45].

**Table 5.** Scenario definitions.

| Scenario | Scenario Symbols | Scenario Explanation |
|---|---|---|
| Baseline scenario | A | This scenario is an optimized simulation of the actual situation in 2013. |
| Simulation scenario | B (B1, B2, B3, B4) | This scenario will discuss the changes in farmers' economic behavior in the scenario of loosening land constraints when the TECI increases by 20% (B1), 50% (B2), 80% (B3) and 100% (B4). |
| Policy scenario | C1 | Based on B, this scenario will only restrict land reclamation. |
| | C2 | Based on B, this scenario will only transfer irrigation water to non-agricultural sectors. |
| | C3 | Based on B, this scenario will not only transfer irrigation water to non-agricultural sectors but also restrict land reclamation. |
| Compensation scenario | D1 | Based on C3, this scenario will discuss the situation in which non-agricultural sectors pay for the saved agricultural water at the trade prices of each crop's net income of per unit water. |
| | D2 | Based on C3, this scenario will discuss the impact of raising the proportion of non-agricultural labor to 20%. |

Some assumptions and parameters are set in the different scenarios. On the one hand, the improvement of the TECI is synchronized and dynamic in the same irrigation zone. On the other hand, loosening land constraints means that the unused land can be reclaimed; strengthening land resources management refers to strictly limiting cropland by cultivated area in the statistical yearbook; and strengthening water resources management refers to transferring the saved irrigation water to non-agricultural sectors. Furthermore, non-agricultural sectors pay compensation to farmers through purchasing the saved irrigation water. The tradable water volume refers to the water saved by the improvement of the TECI, and the trade prices refer to each crop's net income of per unit water. Finally, the proportion of non-agricultural labor was approximately 20% in Zhangye City in 2013. However, the proportion of non-agricultural employment in typical irrigation zones is less than 15%. When the saved irrigation water is transferred, the problem of surplus labor will become more prominent. Therefore, this study explores the changes in farmers' behavior when the proportion of non-agricultural employment labor increases to 20%.

### 3.4. Model Calibration and Simulation

The BEM model is constructed by the General Algebraic Modeling System (GAMS) platform [46]. To calibrate the model, the "present situation" is set as the baseline scenario (scenario A) to compare with the actual situation in 2013. The results of the calibrated model are quite consistent with the actual situation in 2013 (Table 6). The average bias of per capital net income of typical irrigation zones

is 4.10%, and the average bias of the cultivated area is only 2.84%. The results are sensitive to the constraints of crop yield, crop rotation, farm-gate prices, changes in animal feed prices and subsidies. The deviation between the simulation results and the actual situation may be partially due to risk attitude and market change. The model assumes that farmers are economically rational and pursue the maximization of net return. Farmers often avoid risk in the actual world. For example, the farmer survey shows that households tend to be reluctant to adopt new agricultural technologies, even though the expected net return is high. In addition, contract farming would affect the results.

**Table 6.** Description of BEM accuracy of typical irrigation zones in Zhangye City.

| Study Areas | Per Capital Net Income (yuan/person) | | | Cultivated Area (hm$^2$) | | | Situation of Land and Water Resources in 2013 |
|---|---|---|---|---|---|---|---|
| | Actual Value | Simulation Value | Bias (%) | Actual Value | Simulation Value | Bias (%) | |
| GDPIZ | 8859 | 8818 | 0.46 | 15,360 | 14,407 | 6.21 | Surplus water resources |
| GYPIZ | 9273 | 9170 | 1.11 | 8322 | 8322 | 0.00 | Water and soil resources are matched |
| GLNDIZ | 8536 | 7739 | 9.34 | 2192 | 2192 | 0.00 | Surplus water resources |
| MYMIZ | 7202 | 7138 | 0.89 | 8697 | 8669 | 0.32 | Inadequate water resources |
| Average bias | 8567.5 | 8216.25 | 4.10 | 8642.75 | 8397.5 | 2.84 | |

Taking the GLNDIZ as an example, this study discusses the operation process of each scenario when the TECI increases by 20%. The scenarios and related parameters are shown in Table 7. The results of each scenario can be obtained by replacing the relevant parameters.

**Table 7.** Changes in parameters in each scenario in the GLNDIZ if the TECI increased by 20%.

| Parameters | A | B (B1) | C1 (B1) | C2 (B1) | C3 (B1) | D1 (B1) [③] | D2 (B1) |
|---|---|---|---|---|---|---|---|
| Irrigation water (m$^3$/hm$^2$) | Cotton: 6485; Seed watermelon: 6570; Maize-wheat: 6615 | | Cotton: 5857; Seed watermelon: 6111; Maize-wheat: 6310 | | | | |
| Total water resources (m$^3$) | 23,173,100 | 23,173,100 | 23,173,100 | 22,643,678 [②] | 22,643,678 | 22,643,678 | 22,643,678 |
| cultivated area (hm$^2$) | 2192 | 63,485 [①] | 2192 | 63,485 | 2192 | 2192 | 2192 |
| Non-agricultural labor (Person) | 924 | 924 | 924 | 924 | 924 | 924 | 1541 |
| Agricultural labor (Person) | 6466 | 6466 | 6466 | 6466 | 6466 | 6466 | 5849 |
| Total labor (Person) | 7705 | 7705 | 7705 | 7705 | 7705 | 7705 | 7705 |

Note: [①] 63,485 = 2192 + 61,293 (unused land area in GLNDIZ); [②] 22,643,824 = 23,173,100−(6485−5857) × 337−(6570−6111) × 289−(6615−6310) × 607; [③] The results are achieved by adjusting the objective function.

## 4. Results

Changes in farmers' economic behavior in the simulation, policy, and compensation scenarios are obtained based on the scenario A. The shadow price of main resources derived from the BEM model can reflect the scarcity of main resources and explain changes in farmers' economic behavior (Table 8). The results reveal that water resources are the key constraint on agricultural production, and transferring the saved agricultural water will have a significant impact on farmers' economic behavior in the GDPIZ, GYPIZ, and MYMIZ. However, in the GLNDIZ, water resources are not the constraint on agricultural production. The changes in available land resources will have a greater

impact on farmers' economic behavior, because the shadow price of land resources is about 5-times the current situation [31]. Furthermore, the agricultural labor is obviously not a limiting factor in the study area. The impact of capital on farmers' economic behavior is relatively small. Compared with the farmer survey in 2013, the scarcity of resources reflected by the shadow price is consistent with the current situation of the study area.

**Table 8.** The shadow prices of relevant resources under the actual situation and the scenario A of typical irrigation zones in Zhangye City.

| The Shadow Prices of Relevant Resources | GDPIZ | | GYPIZ | | GLNDIZ | | MYMIZ | |
|---|---|---|---|---|---|---|---|---|
| | Actual Situation | Scenario A | Actual Situation | Scenario A | Actual Situation | Scenario A | Actual Situation | Scenario A |
| Water resources (yuan/m$^3$) ① | 0.12 | 1.26 | 0.12 | 1.85 | 0.12 | 0.12 | 0.12 | 3.03 |
| Land resources (yuan/hm$^2$) ② | 7785 | 0 | 7800 | 1737.09 | 3000 | 16,464.08 | 6420 | 0 |
| Agricultural labor (yuan/person) ③ | 100 | April: 56.10; July: 12.78; Other months are zero. | 130 | April and July: 12.78; August: 7.83; Other months are zero. | 100 | June: 17.43; August: 22.33; Other months are zero. | 80 | March: 12.43 April: 11.21; August: 0.46; Other months are zero. |
| Capital ④ | 0.07 | 0 | 0.08 | 0 | 0.08 | 0 | 0.07 | 0.243 |

Note: ① The price of irrigation water is based on the farmer survey; ② The prices of land resources is expressed by the average of farmer's land transfer fee; ③ The price of agricultural labor is expressed by the average cost of hiring daily labor; ④ The price of capital is expressed by interest.

## 4.1. Changes in Agricultural Water Consumption and Farmers' Land-Expansion Behavior

The changes in the cropland area of the typical irrigation zone are presented in Table 9. The results show that in the case of loosening land constraints, farmers will reuse all the saved water for agricultural production by reclaiming unused land or increasing the area of water-intensive crops (vegetables) with the improved TECI. For example, in the GDPIZ, compared with scenario A, if the TECI increases by 20%, the cropland area will decrease by 6.21%, and if the TECI continues to increase by 50%, the cropland area will increase by 27.78%. After that, the cropland area will increase at a decreasing rate. In the end, if the TECI increases by 100%, the cropland area will expand by 51.91%, from 14,407 hm$^2$ to 21,884 hm$^2$. In the GLNDIZ, when the TECI increases by 80%, the cropland area will expand by a maximum of 36.69%. After that, the cropland area will increase at a decreasing rate. When the TECI increases by 100%, the cropland area will expand by 50.66%, from 2192 hm$^2$ to 3302 hm$^2$. However, in the MYMIZ, when the TECI increases by 80%, the cropland area will expand by a maximum of 17.42%, and then it will decrease. However, in the GYPIZ, if the land constraints are loosened, the cropland area will decrease by 22.66% because farmers will grow more water-intensive vegetables, and the saved irrigation water will be insufficient to permit the expansion of cultivated land. These shows that with the increase of available water, the binding force of water on farmers' land-expansion behavior will gradually decline in the irrigation with inadequate water, and farmers will strengthen the reclamation of unused land. While in the irrigation zones with surplus water, farmers can reclaim more unused land because of the decline in the binding force of land resources in scenario B.

In scenario C1, the cropland area of all irrigation zones is the same as that in scenario A. In scenario C2, the results show that transferring the saved irrigation water to non-agricultural sectors can significantly reduce land expansion. Compared with scenario B, the land-expansion area will decrease greatly, indicating that strengthening water resources management is the key to controlling land-expansion. The cropland area will be lower than the actual area in irrigation zones with inadequate water. For example, in the MYMIZ, farmers will abandon farming if the water resources cannot support the scale of agricultural production. Therefore, we must pay attention to the farmers' abandonment of cultivated land and increase employment opportunities for surplus

agricultural labor. However, in irrigation zones with surplus water, such as GLNDIZ, farmers will continue to expand cultivated land. Therefore, farmers' land-expansion behavior can be largely restricted by transferring the saved irrigation water to non-agricultural sectors in irrigation zones with inadequate water, but to contain land expansion in the irrigation zones with surplus water, the policy of restricting land reclamation must be implemented simultaneously. In addition, in scenarios D1 and D2, the farmers' cultivated area is the same as that in scenario C3.

**Table 9.** Changes in the cropland area in different scenarios of typical irrigation zones Unit: hm$^2$.

| Study Areas | Scenarios | B1 | B2 | B3 | B4 |
|---|---|---|---|---|---|
| GDPIZ | A | 14,407 | | | |
| | B | 13,513 | 18,408 | 20,348 | 21,884 |
| | C1 | 15,153 | 15,360 | 15,360 | 15,360 |
| | C2 | 13,065 | 15,868 | 15,854 | 15,844 |
| | C3 | 14,224 | 15,360 | 15,360 | 15,360 |
| GYPIZ | A | 8322 | | | |
| | B | 6343 | 6343 | 6343 | 6343 |
| | C1 | 8322 | 8322 | 8322 | 8322 |
| | C2 | 6318 | 6280 | 6243 | 6218 |
| | C3 | 8322 | 8322 | 8322 | 8322 |
| GLNDIZ | A | 2192 | | | |
| | B | 2440 | 2564 | 2996 | 3302 |
| | C1 | 2192 | 2192 | 2192 | 2192 |
| | C2 | 2358 | 2350 | 2587 | 2653 |
| | C3 | 2192 | 2192 | 2192 | 2192 |
| MYMIZ | A | 8669 | | | |
| | B | 8866 | 9652 | 10,179 | 9971 |
| | C1 | 8697 | 8697 | 8697 | 8697 |
| | C2 | 8486 | 8185 | 8027 | 7832 |
| | C3 | 8486 | 8185 | 8027 | 7832 |

## 4.2. Changes in Crop-Planting Structure

The different scenarios reveal the changes in the crop-planting structure of typical irrigation zones (Figures 2–5). Compared with scenario A, the area of seed maize and maize will increase, but the proportion of seed maize will be higher in other scenarios in the GDPIZ with the TECI improved. There are two reasons. One is that water resources are the constraint on agricultural production in the GDPIZ, when it becomes the limiting factor, farmers will expand the cultivated area and increase the area of crops that have a higher net income of per unit water. The other is that the water consumption and irrigation cost of seed maize will be reduced with the TECI improved, and the net income of per unit water will be higher than that of maize. Similarly, in the GYPIZ, farmers will choose to grow more field vegetables in scenarios B and C2, and they will slightly increase the area of potato in scenario C3.

In the GLNDIZ, in scenarios B and C2, if the TECI increases by 20% or 50%, the area of food crops, such as maize and wheat, will decrease, while the area of the cash crops, such as tomato, seed watermelon, and cumin, will increase. If the TECI increases by 80%, farmers will begin to grow cotton. The main reason is that in scenario C2, water resources are relatively abundant, and farmers will choose to plant crops with higher net income per unit water. However, in scenarios C1 and C3, the crop-planting structure is the same as in scenario A. The reason is that the water resources in the GLNDIZ are surplus, the transfer of saved irrigation water does not make water a constraint, and the crop-planting structure remains unchanged when the land resources are restricted.

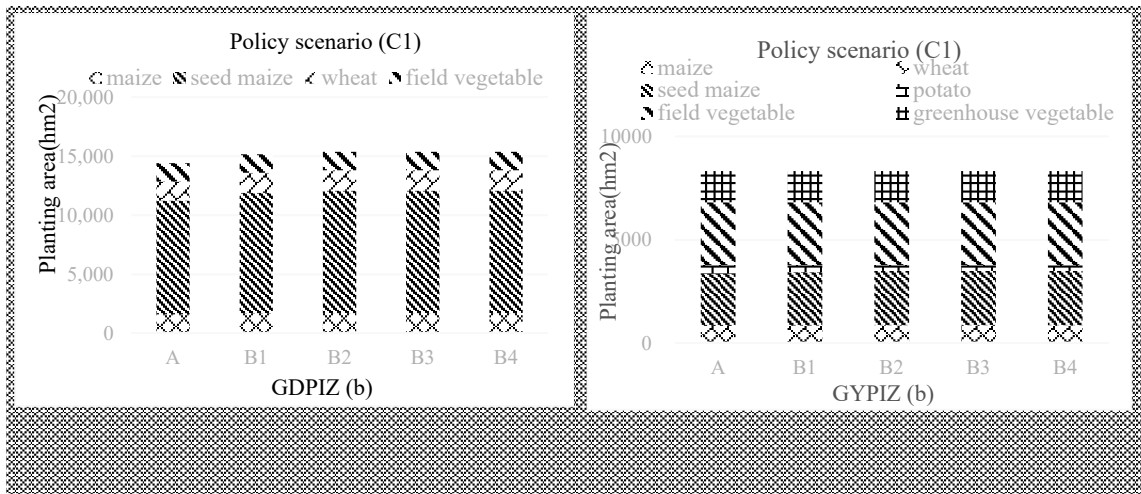

**Figure 2.** Changes in the crop-planting structure in scenario B of typical irrigation zones; (**a**) in the GDPIZ; (**b**) in the GYPIZ; (**c**) in the GLNDIZ; (**d**) in the MYMIZ.

**Figure 3.** *Cont.*

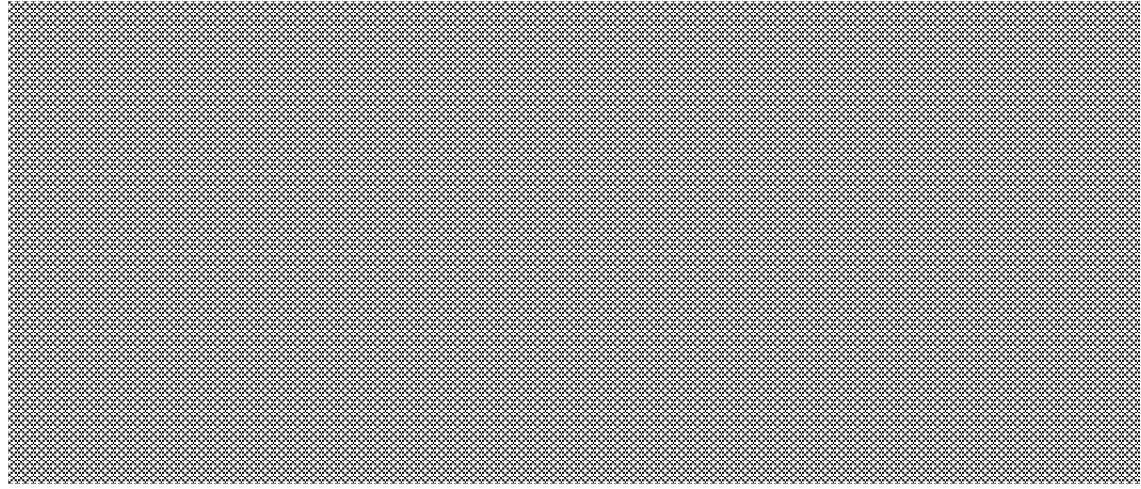

**Figure 3.** Changes in the crop-planting structure in scenario C1 of typical irrigation zones; (**a**) in the GDPIZ; (**b**) in the GYPIZ; (**c**) in the GLNDIZ; (**d**) in the MYMIZ.

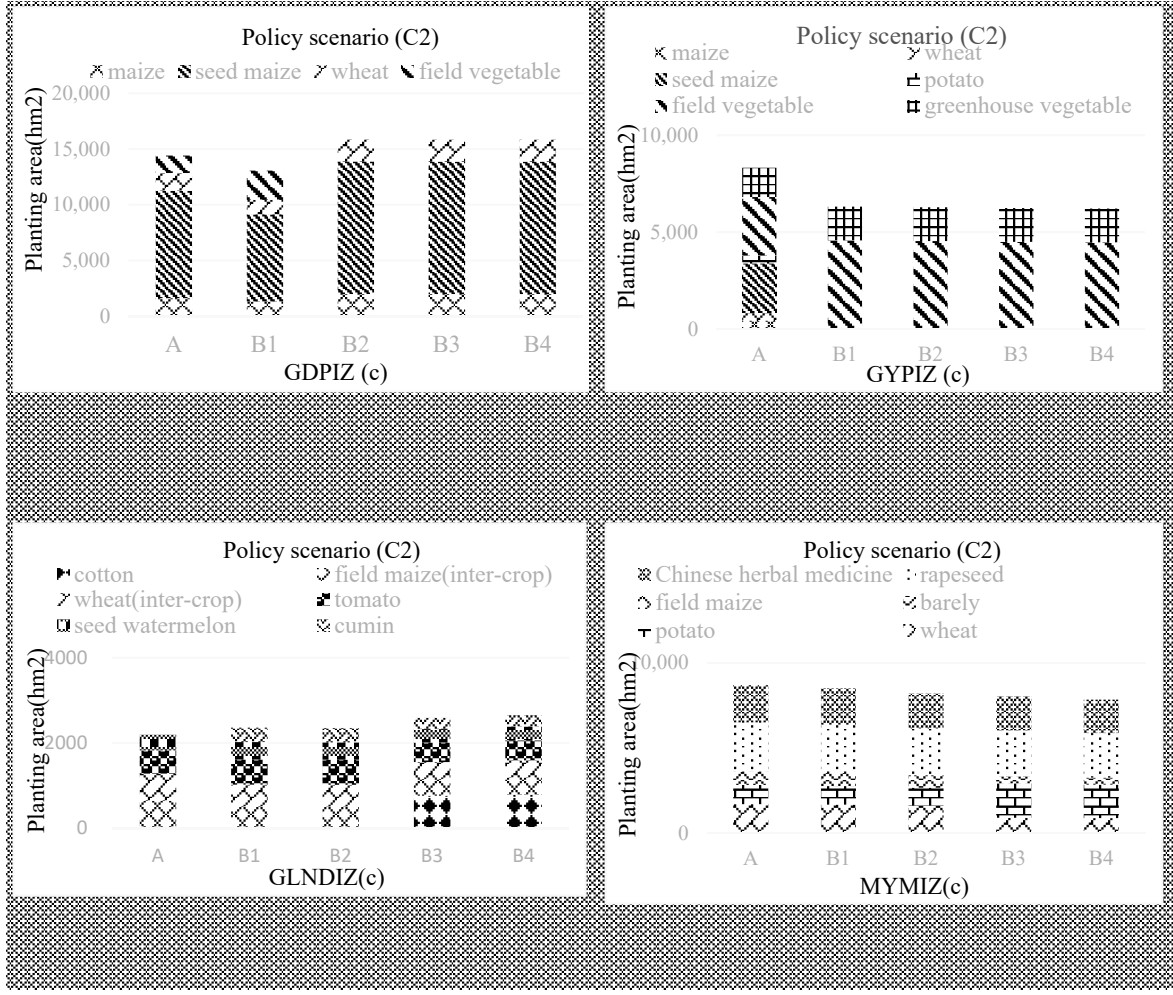

**Figure 4.** Changes in the crop-planting structure in scenario C2 of typical irrigation zones; (**a**) in the GDPIZ; (**b**) in the GYPIZ; (**c**) in the GLNDIZ; (**d**) in the MYMIZ.

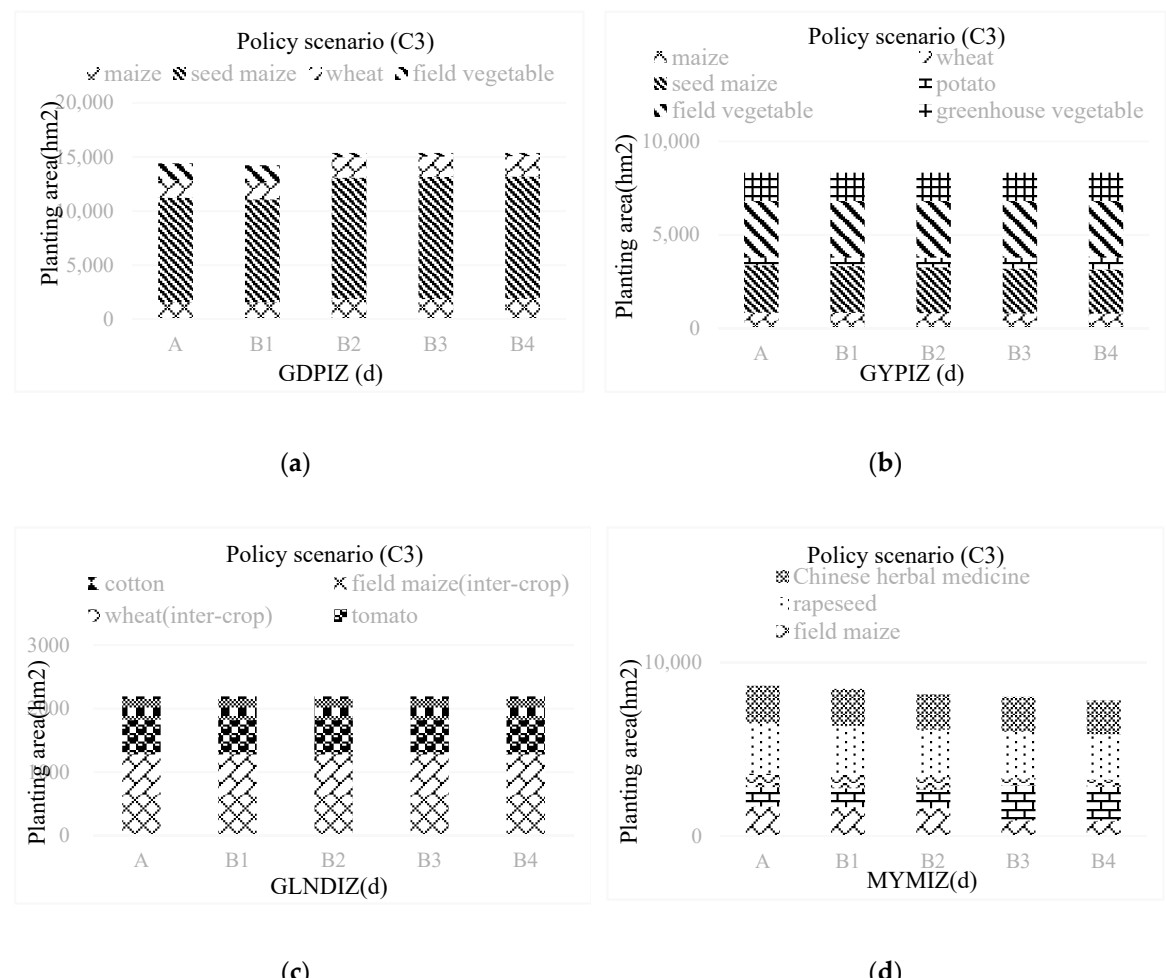

**Figure 5.** Changes of the crop-planting structure in scenario C3 of typical irrigation zones; (**a**) in the GDPIZ; (**b**) in the GYPIZ; (**c**) in the GLNDIZ; (**d**) in the MYMIZ.

In the MYMIZ, farmers will choose to increase the area of cash crops, such as potato, Chinese medicinal materials, and rapeseed, but reduce the area of food crops, such as wheat, in scenarios B, C2, and C3. Overall, the crop-planting structure is dynamically changed in the process of improving the TECI. Farmers will adjust the crop patterns according to the amount of land and water resources and the crop input-output situation.

In addition, the farmers' crop-planting structure in scenarios D1 and D2 is the same as in scenario C3. The reasons are that one is that capital is not a constraint in the GDPIZ, GYPIZ, and GLNDIZ. In the MYMIZ, although the shadow price of water resources is higher than the current price, transferring the saved irrigation water is the main reason leading to changes in the boundaries of agricultural production constraints. If the amount of water available is reduced, farmers' economic behavior will change significantly. The other is that the agricultural labor is not a constraint in the study area, so raising the proportion of non-agricultural labor to 20% does not change farmers' crop-planting structure and land-expansion behavior.

### 4.3. Effects on Farmers' Economic Income

The changes in farmers' economic income are shown in Table 10. Compared with scenario A, due to the land expansion and the reduction of irrigation cost, the per capita net income of all irrigation zones increases by approximately 5% in both scenarios B and C1. However, the per capita net income of most irrigation zones can be decreased or at least not increased with the transfer of agricultural water in scenarios C2 and C3. The main reason is that the reduction of irrigation water production

factors caused by transferring the saved irrigation water to non-agricultural sectors hinders rural economic development.

**Table 10.** Changes in the per capita net income in different scenarios of typical irrigation zones Unit: yuan/person.

| Study Areas | Scenarios | B1 | B2 | B3 | B4 |
|---|---|---|---|---|---|
| GDPIZ | A | | 8818 | | |
| | B | 8939 | 9122 | 9390 | 9603 |
| | C1 | 8921 | 8962 | 8977 | 8987 |
| | C2 | 8795 | 8777 | 8793 | 8803 |
| | C3 | 8804 | 8793 | 8777 | 8782 |
| | D1 | 9060 | 9433 | 9546 | 9742 |
| | D2 | 8847 | 8933 | 8939 | 8943 |
| GYPIZ | A | | 9179 | | |
| | B | 9538 | 9538 | 9538 | 9538 |
| | C1 | 9177 | 9184 | 9184 | 9185 |
| | C2 | 9521 | 9496 | 9471 | 9454 |
| | C3 | 9166 | 9161 | 9154 | 9150 |
| | D1 | 9175 | 9181 | 9187 | 9191 |
| | D2 | 9618 | 9612 | 9606 | 9602 |
| GLNDIZ | A | | 7740 | | |
| | B | 7963 | 8108 | 8380 | 8857 |
| | C1 | 7745 | 7736 | 7744 | 7759 |
| | C2 | 7846 | 7843 | 7918 | 8000 |
| | C3 | 7745 | 7736 | 7744 | 8000 |
| | D1 | 7893 | 8107 | 8511 | 8338 |
| | D2 | 8320 | 8328 | 9615 | 8336 |
| MYMIZ | A | | 7138 | | |
| | B | 7182 | 7249 | 7328 | 7389 |
| | C1 | 7147 | 7150 | 7153 | 7155 |
| | C2 | 7102 | 7042 | 6983 | 6945 |
| | C3 | 7102 | 7042 | 6983 | 6945 |
| | D1 | 7242 | 7297 | 7392 | 7455 |
| | D2 | 8233 | 8172 | 8114 | 8076 |

In scenarios D1 and D2, if the non-agricultural sectors pay for the saved agricultural water at the trade prices of each crop's net income of per unit water, farmers' economic loss can be compensated. In addition, if the government raises the proportion of non-agricultural labor to 20%, the increase in non-agricultural income can also make up for the farmers' economic loss. Therefore, it is necessary to improve water rights trading and increase employment opportunities for surplus agricultural labor to promote economic development in rural areas.

## 5. Discussions

### 5.1. Driving Mechanism of Farmers' Economic Behavior

In scenario B, farmers will expand cultivated land and increase the area of cash crops, and these behavior leads to water rebound effects. In scenarios C2 and C3, farmers' land-expansion behavior is greatly restricted by transferring the saved irrigation water. The changes in farmers' economic behavior are affected by internal and external factors. On the one hand, the internal factors mainly refer to farmers' pursuit of maximum profit. Farmers' production decisions are determined considering input and output. The improvement of the TECI can reduce the water consumption and irrigation cost. The movement of crop input-output balance points will lead to changes in farmers' economic behavior, and the BEM model can describe these results. On the other hand, the external factors mainly refer to the constraints of water and land resources. Farmers' production decisions will change by adjusting

the constrained boundary of water and land resources to achieve the goal of maximizing income. When land constraints are loosened, water resources will become the limiting factor, and farmers will expand the cultivated area and increase the area of crops that have a higher net income of per unit water. When land resources become the limiting factor, farmers will increase the area of crops that have a higher net yield of per acre. Therefore, changes in farmers' behavior are mainly determined by internal factors, but they also need to be adjusted according to changes in external conditions.

## 5.2. Avoiding the Water Rebound Effect

The existence of water rebound has seriously reduced the effectiveness of water resources management policy. Many researchers have analyzed the effects of more efficient irrigation using theoretical model simulation or empirical comparative analysis and shown that efficiency improvements do not always reduce overall water use [47]. For example, Ward et al. (2008) revealed that water use unexpectedly reduced and water depictions actually increased in the Upper Rio Grande Basin of North America [3], Lopez-Gunn et al. (2012) showed that real saving was less than theoretical saving in the Alicante and Almería in Spain [48], Fernandez Garcia et al. (2014) indicated that water diverted for irrigation reduced but irrigation water demand increased in five irrigation zones of Andalusia [5], and Wu et al. (2018) explained that high-efficiency irrigation technologies clearly contributed to the decrease of agricultural water but induced water rebound in total water use in the HRB [49], and so on.

Unlike previous researchers who empirically investigated the water rebound from the aspects of connotation, influence, and rebound degree, this study extend the past work as follows: firstly, the study provides the amount of saved irrigation water based on the results of the TECI, which suggest that the key issue is how to measure the amount of saved agricultural water. The irrigation quota may not be the actual water demand of crops; we should improve the monitoring system to obtain the actual water demand of crops and strictly transfer the saved irrigation water to non-agricultural sectors to avoid water rebound; Secondly, it simulates the implementation effect of unified water resources management policy from the perspective of farmers' behavior. Which indicated that to restrict farmers' land-expansion behavior, it is necessary to formulate policies according to local conditions. For example, it is necessary to implement a policy of transferring agricultural water and restricting land reclamation simultaneously in regions with surplus water, but in regions with inadequate water, only the policy of transferring agricultural water needs to be implemented. These results may provide scientific reference for avoiding water rebound in the HRB and areas with similar problems.

## 5.3. Alleviating Water Use Conflicts by Improving the Water Rights Trading Systems

Water rights trading systems are becoming an important way to alleviate water use conflicts and achieve distributive efficiency for water resources. Many studies have revealed that the transaction between farmers was only likely to occur where surplus water exist and farmers were unlikely to explore the opportunity cost of using water in the immediate future [24] and tiered prices based on the value of each crop's marginal product have recently been suggested as an efficient pricing method [25]. In addition, the content of these studies was mainly focused on third-party effects [50], potential economic gains [51], barriers or transaction costs [52]. However, few studies explored the changes in farmers' economic behavior under these policies. This study simulates the impact of cross-sector water rights trading on farmers' economic behavior based on the TECI and each crop's net income of per unit water in scenario D1, which makes up for the lack of relevant research and provides reference for the improvement of water rights trading systems. When reducing agricultural water use becomes an inevitable choice for promoting the sustainable development of oasis regions, the water rights trading systems must be improved to ensure the smooth transfer of agricultural water. Finally, some suggestions are put forward for improving the water rights trading systems. First, the government should abandon the policy of reducing farmers' water quotas gradually and transferring the saved irrigation water to non-agricultural sectors via market-oriented channels. Second, we should accurately

provide tradable water volume for water rights trading by upgrading the technical systems. Third, we can alleviate the social obstacles to water rights trading by increasing household non-agricultural income [53].

## 6. Conclusions

This study explores farmers' economic behavior in response to the policy of transferring irrigation water and restricting land reclamation with the improved TECI in the midstream of the HRB by developing the BEM model. To better reflect the scarcity of main resources in different irrigation zones and explain changes in farmers' economic behavior in different scenarios, this study provides the shadow price of the main resources and the main findings are summarized as follows: first, if the land constraints are loosened and the saved irrigation water is not transferred, farmers will reuse all of the saved water for agricultural production by reclaiming unused land or increasing the area of water-intensive crops. In scenario B, farmers will expand the cropland area in most irrigation zones, but in the GYPIZ, farmers will increase the area of field vegetables and greenhouse vegetables, which are the most water-intensive crops. These behaviors lead to water rebound. In scenario C1, although the policy of restricting land reclamation can restrict land expansion, it cannot avoid water rebound caused by adjusting the crop-planting structure. In scenarios C2 and C3, farmers' land-expansion behavior can be largely restricted by transferring the saved irrigation water to non-agricultural sectors in irrigation zones with inadequate water, but to contain land-expansion behavior in the irrigation zones with surplus water, the policy of restricting land reclamation needs to be implemented simultaneously. Second, because water resources become a limiting factor in scenarios B, C2, and C3, farmers will choose to increase the area of cash crops, such as seed maize, vegetables, tomato, seed watermelon, potatoes and rapeseed, and reduce the area of food crops, such as wheat, and maize. Farmers will adjust the crop patterns according to the amount of land and water resources and the crop's input-output situation, and the crop-planting structure will change dynamically in the process of improving the TECI. Finally, farmers' economic income can be decreased or at least not increased in scenarios C2 and C3. Both benefit compensation from non-agricultural sectors and raising non-agricultural income can compensate farmers' economic loss. In summary, transferring the saved irrigation water is one of the most effective policies to avoid the negative effects of high-efficiency irrigation technologies, and the improvement in water rights trading and the migration of surplus labor can reduce the adverse impact on farmers' production.

The coordinated development of the agricultural economy and the eco-environment is an enduring topic. In the past 20 years, the implementation of the water reallocation plan and the building of a water-saving society have not reduced agricultural water use in the HRB. This situation is predicated on the provision of sufficient water resources. According to official data of the Heihe River Bureau of the Yellow River Conservancy Commission, the HRB has been rich in water resources since the 21st century. However, when it enters a period of insufficient water resources, the agriculture water demand will be difficult to meet, which will lead to more serious water conflicts. To accelerate the building of a water-saving society, we should prevent problems from arising by transferring saved agricultural water and improving water rights trading.

**Author Contributions:** G.L. conducted the BEM model simulations and data analysis, wrote and revised the paper. D.Z. provided academic advice throughout the process and helped to revise the manuscript. M.S. suggested the research theme and provided expert advice throughout the paper, and revised the paper.

**Funding:** This research was funded by National Natural Science Foundation of China (No. 91325302).

**Acknowledgments:** The authors would like to thank the participants of the project, such as Xiaojun Wang from the University of Chinese Academy of Sciences, Xiao Li from the Beijing Normal University, etc., for their contributions on farmer survey.

**Conflicts of Interest:** The authors declare no conflicts of interest.

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
