# Peer review of "How Do Farmers Respond to Water Resources Management Policy in the Heihe River Basin of China?"

_sustainability, doi:10.3390/su11072096_

Round 1
Reviewer 1 Report
@page { margin: 2cm } p { margin-bottom: 0.25cm; line-height: 120% }
The topic is highly relevant, and the objective of the research is clearly stated. The methodology is not innovative, but its application to water management is relevant for the scientific community.
It is the first time I read about the terms “compressed agricultural water”. I checked their meaning, not only within the scientific references, but also in every-day information records, but I did not find them. I recommend that you find some alternative terms, such as “limitation of water quantities”,or similar (please consult a native English speaker for the proper terms to be used in this context).
Between lines 49-53(page 2) you describe the so-called “Jevons paradox” or “Jevons effect”, occurring when efficency gains conceived to reduce the resource consumption, in fact will determine an opposite effect, due to the reduction of relative cost for the resource use. Please, mention this phenomenon in the paper, with the appropriate references referred to water management.
At line 153 (page 3) you should specify to whom the net income is referred to; i guess it is referred only to farmers’income.
At line 214-215 you mentioned about the calibration of the model to generate the “baseline scenario”. It is very important to specify in detail how you performed the calibration of the model. This is important to grasp the expected range of variation of the output one can expect from the simulation of other scenarios. In case you adopt a set of constraints, we cannot expect large changes. In addition, there is another way to verify the reliability of the GAMS model, that is by reporting the shadow prices of the main resources. In this case, please provide the shadow prices of land, labor, capital and water, and discuss if they are consistent within the current economic situation of the study area.
From the scientific point of view, this is the most important limit of the current paper. But I am confident that the Authors are able to improve it.
Across the paper you never mention about the need for metering water. Are there already working efficient water metering systems in the study area? How information of water quantity is already used to monitoring & control of water resources at farm level? How effective is the enforcement of the current (or, eventually, the future) water regulation? In the discussion, you should put some arguments related to the actual feasibility of some scenarios, compared to other.
Author Response
Response to Reviewer 1 Comments
Point 1: It is the first time I read about the terms “compressed agricultural water”. I checked their meaning, not only within the scientific references, but also in every-day information records, but I did not find them. I recommend that you find some alternative terms, such as “limitation of water quantities”,or similar (please consult a native English speaker for the proper terms to be used in this context).
Response 1: Thanks to the reviewer for the comments. The author will further improve the ability of English expression. The term “compressed agricultural water” appeared twice in the paper, at line 1 (page 1) and 380 (page 13). According to reviewers' comments, the authors have carefully reviewed the literature and revised the “compressed agricultural water” to “reducing agricultural water use”, at line 1 (page 1) and 384 (page 13).
Point 2: Between lines 49-53(page 2) you describe the so-called “Jevons paradox” or “Jevons effect”, occurring when efficiency gains conceived to reduce the resource consumption, in fact will determine an opposite effect, due to the reduction of relative cost for the resource use. Please, mention this phenomenon in the paper, with the appropriate references referred to water management.
Response 2: Thanks for this comment. We studied some literature on water rebound effect and reorganized the literature on the application of "Jevens Paradox" in water resources management. References 9-11, 14 and 47-51 give the authors great inspiration. The authors have adopted this comment and added the "Jevens Paradox" or “Khazzoom-Brookes” hypothesis between lines 53-55 (page 2). What’s more, we have summarized the contents of these references in detail between lines 346-357 (page 13).
Point 3: At line 153 (page 3) you should specify to whom the net income is referred to; i guess it is referred only to farmers’ income.
Response 3: According to the reviewer's comments, we found that the previous expression was not accurate enough. So we have adopted the comments and revised the “net income” to the “farmers’ net income” between lines 159-160 (page 5).
Point 4: At line 214-215 you mentioned about the calibration of the model to generate the “baseline scenario”. It is very important to specify in detail how you performed the calibration of the model. This is important to grasp the expected range of variation of the output one can expect from the simulation of other scenarios. In case you adopt a set of constraints, we cannot expect large changes. In addition, there is another way to verify the reliability of the GAMS model. That is by reporting the shadow prices of the main resources. In this case, please provide the shadow prices of land, labor, capital and water, and discuss if they are consistent within the current economic situation of the study area.
Response 4: This comment has an important reference value for us to further improve the model. To explain this problem, we have divided the contents of 3.3. into two parts, namely 3.3. scenario design and 3.4. Model calibration and simulation. The BEM model calibration mainly referred to the verification of model parameters in the paper. The accuracy of the model was improved by comparing the main parameters of the “baseline scenario” with the actual situation in 2013. Two main parameters of “per capital net income” and “cultivated area” were proofread, because the main line of this paper was the farmers’ economic behaviour. Referring to the reviewer's comments, the authors have further calculated the average bias of the main parameters of the typical irrigation zones in Table 6 (page 8). The results showed that the average bias of per capital net income was 4.10% and the average bias of the cultivated area was only 2.84% (between lines 232-233). This indicated that the BEM model had higher accuracy and the simulation results were relatively reliable. In addition, we also pointed out that the model results were sensitive to the constraints of crop yield, crop rotation, farm-gate prices, changes in animal feed prices and subsidies (between lines 233-234).
In addition, thanks to the reviewer for the issue of model constraints, we carefully considered it. The choice of the constraints of resource endowment, which included land area, family labour, livestock feed requirement, nutritional requirement, capital, water resources and crop rotation, was mainly based on the the actual situation of typical irrigation zones in 2013. And in order to further verify the accuracy and sensitivity of the model, our team revisited the typical irrigation zones in August 2017 and proofread the main parameters of the baseline scenario.
What's more, the authors have repeatedly discussed the issue of shadow prices. In many studies, the shadow prices were often used to reflect the scarcity of resources and the marginal benefit of resource use. The BEM model can be used to examine the shadow prices, for example, Shi et al.(2014) (Refs. 31) calculated the shadow price of water resource to provide advice for water resources management policy in the Heihe River Basin. But few studies have verified the reliability of BEM models by calculating shadow prices. We are very grateful to the reviewer’s comments. These valuable suggestions will definitely provide important reference for our future study.
Point 5: Across the paper you never mention about the need for metering water. Are there already working efficient water metering systems in the study area? How information of water quantity is already used to monitoring & control of water resources at farm level? How effective is the enforcement of the current (or, eventually, the future) water regulation? In the discussion, you should put some arguments related to the actual feasibility of some scenarios, compared to other.
Response 5: According to reviewers' comments, the authors have carefully reviewed the literature on water resources management in the Heihe River Basin (Refs. 8, 29-32, etc.) and summarized the contents of these references in detail between lines 96-102 (page 3). The main findings were that a water quota was a more suitable choice for the purpose of reducing agricultural water use and many studies focused on encouraging farmers to actively participate in water resources management. These provided an important reference value for this paper.

Reviewer 2 Report
The paper is well written. Just minor edits (e.g. avoid the use of "we" etc. - see attached document). The discussions and conclusions are a bit difficult to follow. This should be restructured. The journal is for an international reach so it is important to show how the methods used and the results could be used in similar environments (e.g. in counties and countries with similar water use policies). What is the innovation with regards to the research done - what does it contribute to the scientific community? These are important to address.

Author Response
Response to Reviewer 2 Comments
Point 1: The paper is well written. Just minor edits (e.g. avoid the use of "we" etc. - see attached document). The discussions and conclusions are a bit difficult to follow. This should be restructured. The journal is for an international reach so it is important to show how the methods used and the results could be used in similar environments (e.g. in counties and countries with similar water use policies). What is the innovation with regards to the research done - what does it contribute to the scientific community? These are important to address.
Response 1: Thanks to the reviewer for the comments. Referring to the reviewer’s comments, the authors have seriously considered the differences between the discussions and conclusions and revised as follows:
On the one hand, the authors have reorganized the literature on water rebound effect (Refs. 9-11, 14, 47-51, etc) and summarized the contents of these references in detail between lines 346-357 (page 13). The main findings were that water rebound was widespread and not well managed in many countries/regions, such as the Upper Rio Grande Basin of North America, the Alicante and Almería in Spain, irrigation zones of Andalusia, the HRB of China and so on. Unlike previous studies, the contribution of this paper not only provided the amount of saved irrigation water based on the results of the TECI, but also simulated the implementation effect of unified water resources management policy from the perspective of farmers’ behaviour between lines 358-371 (page 13).
On the other hand, the authors have summarized the literature on water rights trading and water use conflicts (Refs. 52-57, etc.) and pointed out the differences between lines 372-383 (page 13).
Finally, the authors have revised the vocabulary expressed in the first person.

Reviewer 3 Report
this paper introduces us to the impact of water resources within agriculture processes in China, and shows the trends in farmers.
i would appreciate a more extended introduction, especially in the topics of water use conflict in agriculture
the existing literature review is adequate, therfeore i would appreciate an enlargement.
Author Response
Response to Reviewer 3 Comments
Point 1: This paper introduces us to the impact of water resources within agriculture processes in China, and shows the trends in farmers. I would appreciate a more extended introduction, especially in the topics of water use conflict in agriculture. The existing literature review is adequate, therefore i would appreciate an enlargement.
Response 1: Thanks to the reviewer for the comments. According to reviewers' comments, the authors have carefully reviewed the literature on water resources management (Refs. 8, 29-32, etc.), water use conflict and water rights trading (Refs. 52-57, etc.) and water rebound effect (Refs. 9-11, 14 and 47-51). The contents of lines 96-102 (page 3), 372-383 (page 13) and 346-357 (page 13) have been revised in detail.

Round 2
Reviewer 1 Report
The paper has been improved remarkably, with respect to its previous version. Authors made strong effort to reply to comments and suggestions.
However, two relevant issues are still unsolved, which refers to former question n.4 "At line 214-215 you mentioned about the calibration of the model to generate the “baseline scenario”. It is very important to specify in detail how you performed the calibrationof the model. This is important to grasp the expected range of variation of the output one can expect from the simulation of other scenarios. In case you adopt a set of constraints, we cannot expect large changes. In addition, there is another way to verify the reliability of the GAMS model. That is by reporting the shadow prices of the main resources. In this case, please provide the shadow prices of land, labor, capital and water, and discuss if they are consistent within the current economic situation of the study area."
In particular, you did note specify which method you adopted to calibrate the model. For instance, you may have added some technical constraints, to pose some upperbound or lowerbound constraints, to certain activities. Or, you may have adjusted some other parameters, in order to find an optimal solution which is similar to the baseline. If you do not clearly specify HOW you performed the calibration, the model is a sort of "black box", which is out of any comprehension for the analysts (and the reader).
Besides the values of the parameters you reported in Table 6, shadow prices of most relevant resources will demonstrate how your model is reliable and capable of representing the real world.
If you do not reveal the shadow prices of resources (e.g. land, labor, water, capital), I doubt that your model is reliable. Consequently, the results of simulations cannot be discussed.
The scientific soundness of the model depends only on these critical arguments.
Author Response
Response 1: Thanks to the reviewer for the comments. These comments not only have high theoretical and practical value, but also give the author a great inspiration. Referring to the reviewer’s comments, the authors have seriously reviewed the literature on shadow price and agreed that shadow price of main resources can reflect the scarcity of main resources in typical irrigation zones and explain changes in farmers' economic behaviour in different scenarios.
According to the main content of this paper, the authors have measured the shadow price of water resources, land resources, labor and capital ( Table 8, page 8-9), and discussed whether they are consistent with the current situation of the study area between lines 246-257 ( page 8). The results reveal that water resources is the main constraints of agricultural production in the GDPIZ, GYPIZ and MYMIZ, while land resources is the main constraints in the GLDNIZ. Agricultural labour and capital are not the limiting factors for typical irrigation zones. Compared with the farmer survey in 2013, the scarcity of resources reflected by the shadow price is consistent with the current situation of the study area. Simultaneously, these conclusions also demonstrate the reliability of the baseline scenario.
In addition, according to the results of the shadow price of the main resources, the authors have complemented the relevant content of the results section between lines 278-282 ( page 9) , 302-304 ( page 10) , 322-339 ( page 10) and 420-422 ( page 15).
Thanks again to the reviewer, your comments have greatly improved our paper.

Round 3
Reviewer 1 Report
The requested integration have been included in the paper.
The paper is ready for publication.